# Cryo-EM captures early intermediate steps in dynein activation by LIS1

Kendrick H. V. Nguyen [1], Eva P. Karasmanis [1], Agnieszka A. Kendrick [2], Samara L. Reck-Peterson [1,3,4] ✉ & Andres E. Leschziner [1,5] ✉

Cytoplasmic dynein-1 (dynein) is an essential molecular motor in eukaryotic cells. Dynein primarily exists in an autoinhibited Phi state and requires conformational changes to assemble with its cofactors and form active transport complexes. LIS1, a key dynein regulator, enhances dynein activation and assembly. Using cryo-EM and a human dynein-LIS1 sample incubated with ATP, we map the conformational landscape of dynein activation by LIS1 and identify an early intermediate state that we propose precedes the previously identified dynein-LIS1 Chi state. Mutations that disrupt this species, which we termed "Pre-Chi", lead to motility defects in vitro, emphasizing its functional importance. Together, our findings provide insights into how LIS1 relieves dynein autoinhibition during the activation pathway.

Cytoplasmic dynein-1 is a conserved, minus-end-directed microtubule motor critical for cell division and long-range intracellular transport in eukaryotic cells[1]. Mutations in dynein or its regulators are linked to neurodevelopmental and neurodegenerative diseases in humans[2]. Dynein, a member of the AAA + (ATPase associated with various cellular activities) family, forms a 1.4 MDa dimer composed of two copies each of heavy (HC), intermediate (IC), light intermediate (LIC), and three (TCTEX, LC8, and ROBL) light chains (LC), with the IC, LIC, and LC known as accessory chains (Fig. 1a)[1]. The accessory chains interact with the tail region of the HC domain. For example, the ICs interact with the ROBL, TCTEX, and LC8 LCs, an assembly referred to as the IC-LC tower[3].

Each heavy chain features a motor ring with six AAA + modules, a microtubule-binding domain (MTBD), a stalk connecting the MTBD to the AAA + ring, a buttress extending from AAA5 that contacts the stalk, and a tail with a flexible linker that transmits mechanical forces (Fig. 1a, b)[4–6] Among the six AAA + modules, four (AAA1, AAA2, AAA3, and AAA4) bind ATP, but only three (AAA1, AAA3, AAA4) hydrolyze it[7–11]. The main driver of dynein's mechanochemical cycle is the AAA1 module, which, through ATP hydrolysis, orchestrates the ring's opening and closing, the linker's movement, and the simultaneous rearrangement of the stalk and buttress to regulate the MTBD's affinity for microtubules. The AAA3 and AAA4 modules are regulators of dynein's mechanochemical cycle[12–14].

Cellular and structural studies have shown that dynein primarily exists in a pseudo-twofold symmetric, autoinhibited Phi state incompatible with movement along microtubules[15–17]. Relieving autoinhibition is required for dynein to form a large transport complex comprising one or two dynein dimers (1.4 MDa per dimer), the 1.1 MDa dynactin complex, and an activating adapter, which are cargo-specific and link dynein-dynactin to its cargo. This is known as the Dynein-Dynactin-Activating adapter complex (DDA)[1,18]. The mechanism by which dynein transitions from its autoinhibited state to a fully assembled and active DDA transport complex is not yet fully understood.

LIS1, an essential dynein regulator, is genetically linked to the dynein pathway from fungi to mammals and is mutated in patients with the neurodevelopmental disease lissencephaly[2,19–21]. LIS1 is a dimer (~90 kDa) with an N-terminal dimerization domain (LIS1-N) and C-terminal β-propellers that bind to the dynein motor domain[22–24]. LIS1 plays essential roles in the activation of dynein, including relieving autoinhibition and helping assemble the fully active DDA transport complexes[20,25–32]. Mutations in dynein that block the formation of Phi can partially rescue LIS1 deletion in *S. cerevisiae* and the filamentous fungus *Aspergillus nidulans*, supporting LIS1's role in relieving dynein's Phi-mediated autoinhibition[30–32]. In addition, in vitro assays show that LIS1 enhances the formation of DDA complexes containing two dynein dimers, which move faster than complexes containing a single dynein dimer, supporting its role in active dynein complex assmebly[26–28].

[1]Department of Cellular and Molecular Medicine, University of California San Diego, La Jolla, CA, USA. [2]Salk Institute for Biological Studies, La Jolla, CA, USA. [3]Department of Cell and Developmental Biology, University of California San Diego, La Jolla, CA, USA. [4]Howard Hughes Medical Institute, Chevy Chase, MD, USA. [5]Department of Molecular Biology, University of California San Diego, La Jolla, CA, USA. ✉e-mail: sreckpeterson@ucsd.edu; aleschziner@ucsd.edu

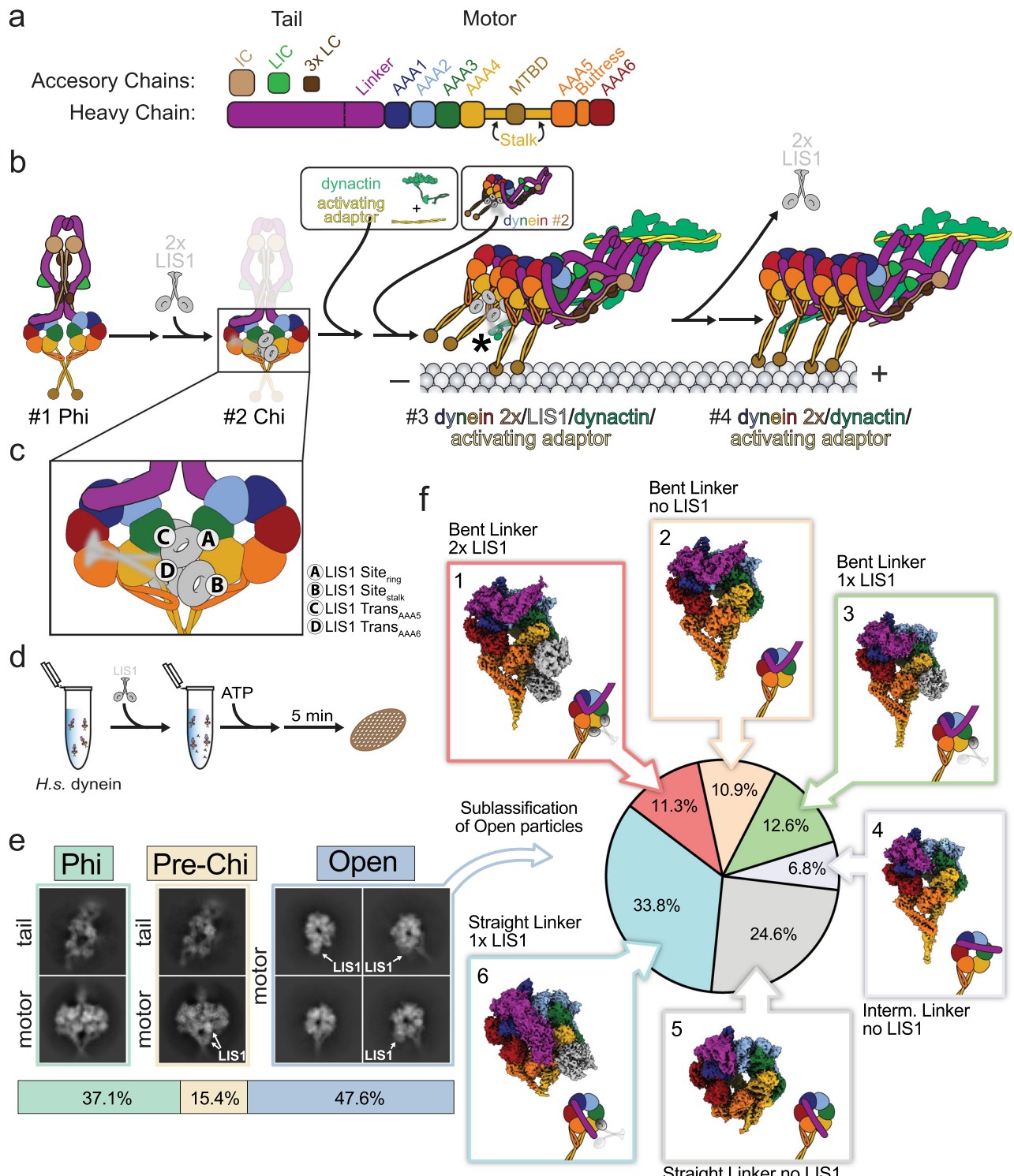

**Fig. 1 | Conformational landscape of dynein activation by LIS1 in the presence of ATP. a** Subunit and domain organization of full-length human dynein. Individual domains and accessory chains (heavy chain (HC), intermediate chain (IC), light intermediate chain (LIC), and three light chains (LC)) are color-coded, and these colors are used throughout the paper. **b** Schematic representation of a hypothetical pathway for dynein activation and assembly by LIS1. The numbers identify species— #1 Phi, #2 Chi, #3 assembly of transport complex, and #4, an active transport complex—that are discussed in the text. The asterisk in #3 indicates a LIS1-p150 dynactin interaction[3], which is also discussed in the text. **c** Known LIS1 binding sites on dynein are shown on the Chi motor domain from panel (**b**). **d** Schematic representation of the cryo-EM sample preparation pipeline. **e** Distribution of particles corresponding to the three main species identified in the cryo-EM dataset: Phi, Pre-Chi, and Open. Representative 2D class averages are shown. In the case of Phi and Pre-Chi, representative 2D class averages of the dynein tails, which were processed separately, are shown above those for the motor domains. LIS1 is indicated whenever present in the averages. **f** Further processing identified six subclasses in the Open species. The particle distribution is indicated with the corresponding cryo-EM maps next to the section in the pie chart.

Given that LIS1 plays multiple roles in the activation and assembly of DDA complexes, several dynein-LIS1 intermediate complexes likely exist. In the current model, dynein activation involves the formation of an intermediate state called Chi, where two LIS1 dimers disrupt the autoinhibited Phi conformation by wedging themselves between the motor domains (Fig. 1b, #1-2)[17,33]. LIS1 promotes the assembly of the Dynein-Dynactin-JIP3-LIS1 (DDA-LIS1) complex by binding to the p150 subunit of dynactin via its LIS1-N domain, while simultaneously maintaining interactions with dynein (Fig. 1b, #3 asterisk)[3]. Once the DDA complex forms, LIS1 likely disengages, allowing the transport complex to move at full velocity (Fig. 1b, #4). LIS1 binds dynein at four sites: between AAA3 and AAA4 in the motor domain (site$_{ring}$), on the stalk (site$_{stalk}$), and, in trans when dynein is in the Chi conformation, at AAA5 (trans$_{AAA5}$) and AAA6 (trans$_{AAA6}$) on a neighboring motor domain (Fig. 1c)[24,27,33-35]. In monomeric dynein, LIS1 has been observed bound to both site$_{ring}$ and site$_{stalk}$[24,34-38], and to site$_{ring}$ in the absence of binding to site$_{stalk}$[27,29,35-38], but not to site$_{stalk}$ only. This is consistent with a model where site$_{stalk}$ binding is weaker and depends on cooperativity with site$_{ring}$.

Although the structure of Chi provided insights into how LIS1 initiates relief of dynein autoinhibition, it was solved using monomeric, truncated yeast dynein motor domains. How autoinhibition is relieved in the context of dimeric, full-length human dynein and how LIS1 enhances this process remains an open question in the field. To address this, we must capture as many states along the dynein activation and assembly pathway as possible. Recent advances in cryo-EM hardware and software have enabled the identification of structural ensembles in samples prepared under native-like reaction conditions[39-45]. In contrast, many previously characterized dynein structures, including Chi, were intentionally trapped in specific states using mutations or non-hydrolyzable ATP analogs. While these approaches provided valuable insights, they limited our ability to observe the full conformational landscape of dynein activation and assembly. To do so requires that we allow dynein to go through its mechanochemical cycle in the presence of ATP, thus populating multiple states. This heterogeneity mining strategy has proven effective in recent studies, uncovering a diverse array of dynein conformations, with and without LIS1[37,46].

Here, we used cryo-EM to visualize the dynamic transitions of full-length human dynein-LIS1 complexes during activation. By incubating full-length dynein with LIS1 and ATP, we captured multiple intermediate states, including a previously uncharacterized dynein-LIS1 complex. This state features partial disruption of the Phi conformation and a distinct set of dynein-LIS1 interfaces. Functional assays demonstrate that these interfaces are essential for dynein activation and high-order assembly in vitro. Together, our findings support a model in which LIS1 facilitates dynein activation by relieving Phi autoinhibition and stabilizing transitional states required for the formation of active transport complexes.

## Results
### Cryo-EM reveals the conformational landscape of full-length human dynein activation by LIS1
To visualize the conformational landscape of dynein activation by LIS1, we incubated full-length human dynein with ATP and LIS1 at room temperature for 5 minutes (Fig. 1d). This sample was vitrified and used to collect cryo-EM data. As expected, data processing revealed significant heterogeneity (Supplementary Figs. 1–3). We identified three major species: individual dynein motors (Open), Phi, and a third state similar to Phi but with two LIS1 β-propellers bound to dynein, which we termed Pre-Chi (Fig. 1e). Two-dimensional (2D) class averages for this state also resembled Chi but lacked the second set of LIS1 β-propellers found in Chi (Fig. 1b #2, c, and e). Based on these and other observations discussed later in the text, we propose that this state represents an intermediate state in the dynein activation pathway.

To fully understand dynein's activation pathway, we began by analyzing particles corresponding to Open. We observed significant conformational and compositional heterogeneity in the Open species and used several approaches to sort out the coexisting subclasses (Supplementary Fig. 2 and Supplementary Movie 1)[39,43,45]. We obtained six different Open subclasses, which we categorized based on the linker conformation (bent, intermediate, or straight) and the presence or absence of LIS1 (Fig. 1f).

For Phi, we obtained a 2.7 Å structure of the motor domain and a 4.4 Å structure of the tail (Supplementary Figs. 1, 3 and 4). The higher resolution of the tail in our map compared to the previously published Phi tail structures[17] allowed us to model additional sections of the dynein heavy and accessory chains (IC, ROBL(LC), and LIC) (Supplementary Fig. 4).

### Pre-Chi may be an early step in dynein's activation pathway
We next turned to the Pre-Chi species. Its 2D classes showed Phi-like features and had only one set of LIS1 β-propellers bound (Fig. 1e and Supplementary Fig. 5) instead of the two seen in Chi. Thus, we named this species Pre-Chi, although we cannot rule out the possibility that this species follows Chi in the dynein activation pathway. We obtained a 3.1 Å structure of the Pre-Chi motor domains (Fig. 2a–d and Supplementary Fig. 1). The nucleotide state of Pre-Chi is the same as that of Phi (Fig. 2e, f and Supplementary Figs. 6, 7). Separately, we obtained a 6.2 Å reconstruction of the tail portion of Pre-Chi (Fig. 2a, b and Supplementary Fig. 3). At the current resolution, we could not observe any differences between the tails of Phi and Pre-Chi (Fig. 3a, b and Supplementary Movie 2). Finally, we analyzed the interface between dynein monomers in Pre-Chi to understand how LIS1 disrupts Phi. The LIS1-induced opening of one side in Pre-Chi disrupts three of the four interfaces involved in Phi stabilization (Linker:Linker, Linker:AAA4, AAA5:AAA5, Fig. 3c–g and Supplementary Movie 3)[17,30]. The Linker:Linker interface is only disrupted on the side where LIS1 is bound (Fig. 3d). The AAA5:AAA5 and Linker:AAA4 interfaces are both weakened, with three of the four salt bridges in Phi being disrupted on the LIS1-bound side in the former (Fig. 3e) and two of three salt bridges disrupted on both faces in the latter (Fig. 3f, g).

Phi dynein has pseudo-two-fold symmetry; the motor domains have true two-fold symmetry, but that symmetry is broken in the tails[17]. We wondered whether the asymmetric nature of the tails biases the binding of LIS1 to one face of the Phi motors by making that side more prone to breathing. Our Pre-Chi structure suggested that LIS1 binds preferentially to the side of Phi where the bulk of the accessory chains of the dynein tail are found (Fig. 2b). To see if we could find a population of Pre-Chi with LIS1 bound to the other face, we revisited the tail processing for Pre-Chi and Phi (Supplementary Fig. 3). We subclassified the Pre-Chi particles and viewed all the volumes (Supplementary Fig. 8). As a point of reference, we looked for density in the volumes that correspond to the IC-LC tower of accessory chains, a feature that is only observable on one face of Phi dynein. Whenever density for the IC-LC tower is present in our Pre-Chi volumes, it is located on the same face as LIS1. We did not observe any volumes with LIS1 bound to the other face (Supplementary Fig. 8b–e).

### Structural comparison between Pre-Chi and Chi
We previously built a model of human Chi based on our structure of yeast Chi to investigate the functional impact of Chi-disrupting mutations on the assembly of the human DDA complex[33]. We compared that model of Chi to Pre-Chi by superimposing one of their heavy chains (Fig. 4a). This alignment showed an increased separation between the motor domains in Chi relative to Pre-Chi, driven by the additional LIS1 dimer wedged between the motor domains in Chi (Fig. 4b and Supplementary Movie 4).

In Chi, the LIS1 bound to the ring of one motor domain (site$_{ring}$) interacts, in trans, with the AAA5 and AAA6 modules of the other

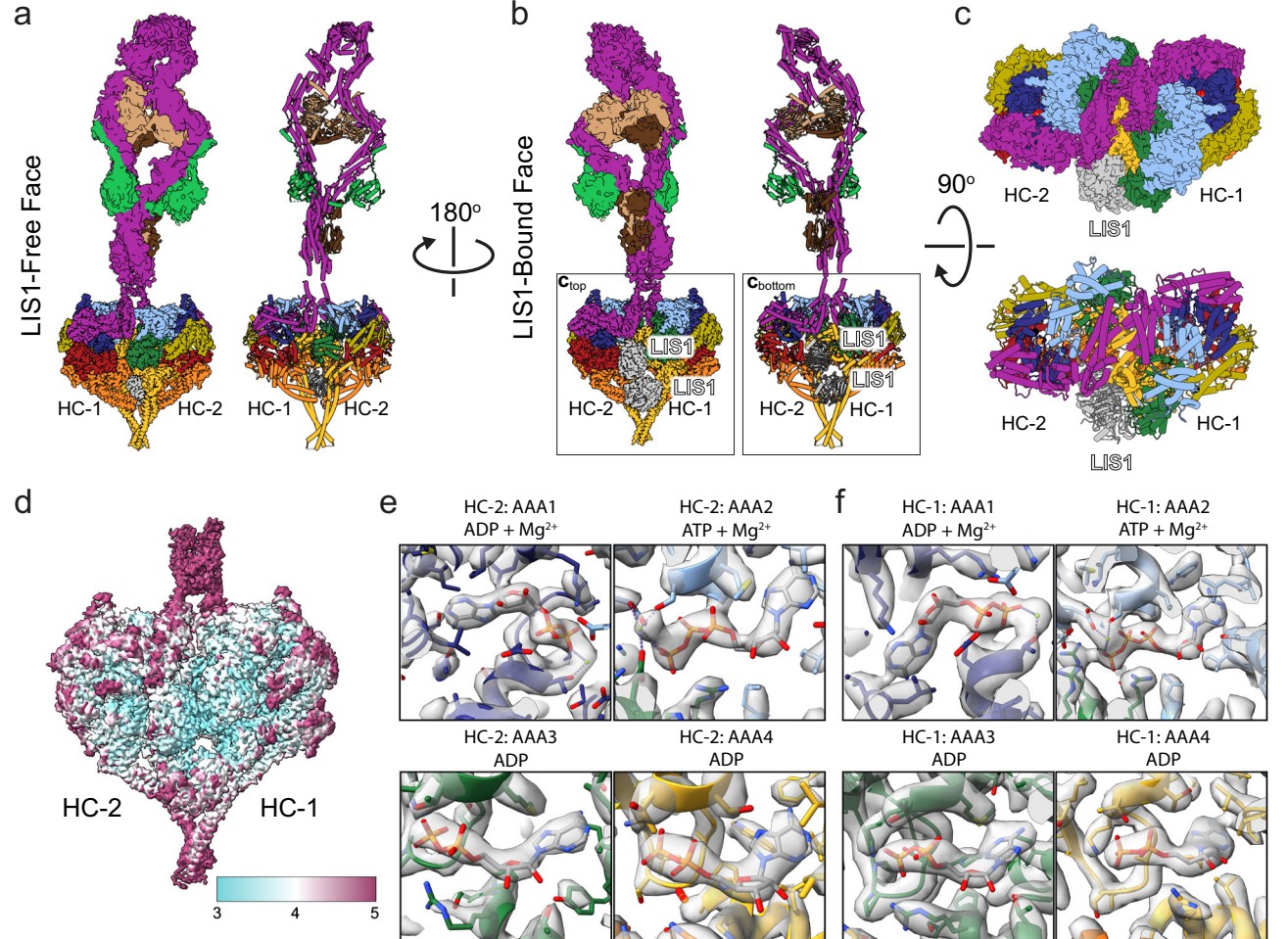

**Fig. 2 | Structure of the Pre-Chi dynein-LIS1 complex. a–c** Cryo-EM maps and models of the motor and tail domains of the Pre-Chi dynein-LIS1 complex are shown in three orientations: (**a**) LIS1-free face, (**b**) LIS1-bound face, and (**c**) a top view where the Pre-Chi motors are shown enlarged and from the perspective of the tail. LIS1 is highlighted in the LIS1-bound (**b**) and top (**c**) views. **d** Local resolution map of Pre-Chi. **e, f** Nucleotide states of AAA1–AAA4 in Motor 2 (Heavy Chain 2, HC-2) (**e**) and Motor 1 (Heavy Chain 1, HC-1) (**f**).

dynein motor (HC-2, Fig. 1c). Based on the yeast Chi model, we predicted that human LIS1 interacts with AAA5 through Y225 and with AAA6 through N203, D205, and D245[33]. In agreement with those predictions, we showed that mutations in these LIS1 residues significantly impaired the formation of activated human DDA complexes[33]. The structure of human Pre-Chi presented here supports our previous modeling (Fig. 4c, d, black circles #1-2, Supplementary Fig. 9a–c, and Supplementary Movie 5).

In addition to the interactions that, based on our models, are common to both Pre-Chi and Chi, we identified two Pre-Chi-specific interfaces (Fig. 4c, d, red circles #3-4, Supplementary Fig. 9a, d, e, and Supplementary Movie 5). The first interface involves a trans interaction between the β-propeller of LIS1 bound to site_ring in one motor domain (HC-1) and the linker domain of the other motor domain (HC-2) (Fig. 4c, d, red circle #3-4 and Supplementary Movie 5). This interaction is mediated by a loop on LIS1 (residues 298–308) that has not been modeled in any structure of human dynein-LIS1 to date[36]. Because this loop is not fully resolved in our map, we could not build individual side chains for residues 300–306, some of which are likely to interact with the linker of HC-2. Interestingly, one of the residues in this region, E300, has been reported to be a missense mutation in lissencephaly patients[47].

The second Pre-Chi interface involves another interaction in trans between LIS1 bound to site_ring in HC-1 and AAA5 of HC-2 (Fig. 4c, d, red circle #4). Notably, the residue involved in this interaction (K3621) is

very close to a residue involved in stabilizing the autoinhibited Phi conformation (E3624).

## The Pre-Chi-specific interface is required for efficient human dynein complex assembly

To assess the functional consequences of disrupting the dynein linker-LIS1 contacts, we identified specifically in Pre-Chi, we turned to in vitro motility assays. In these assays, we measure both the motility and the efficiency of the assembly of active human DDA complexes at the single-molecule level. In vitro, LIS1 enhances the formation of DDA complexes containing two dynein dimers, which move faster than complexes containing a single dynein dimer[26,28]. Our structure showed that dynein's linker contacts LIS1 via a loop (residues 298–308, Fig. 4d red #3). To test the functional role of this contact in Pre-Chi, we deleted this loop in LIS1 (LIS1^{Δ298–308}) and examined the effect of wild-type LIS1 or LIS1^{Δ298–308} on the motility of dynein, dynactin complexes containing the activating adapter BICD2 (DDB) using single-molecule motility assays. As we have previously shown, pre-incubation of DDB with 300 nM wild-type human LIS1 increased the velocity and run frequencies (number of processive runs/ μm of microtubule length) (Fig. 5a–c)[28,33]. In contrast, there was no significant difference in dynein velocity or the run frequencies between DDB complexes alone or those pre-incubated with 300 nM LIS1^{Δ298–308}. These data suggest that the dynein linker–LIS1 contact we identified in our Pre-Chi structure is required for LIS1's role in the efficient formation of active DDB

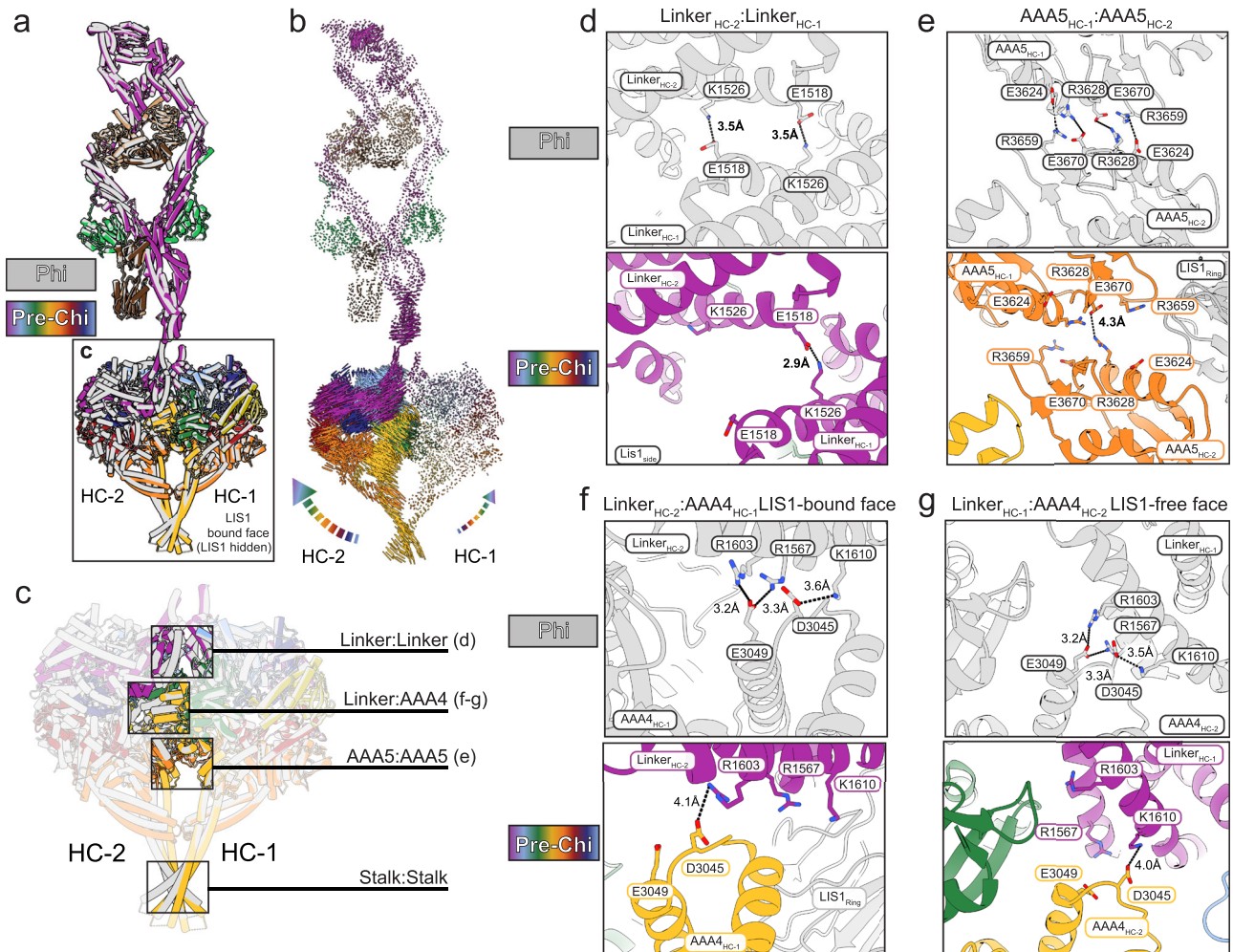

**Fig. 3 | Phi-stabilizing contacts are disrupted in Pre-Chi. a** Superposition of Pre-Chi (rainbow) and Phi (gray) models. HC-1 was used as the reference to align the models. Although the Pre-Chi model is shown from the LIS1-bound face, LIS1 was omitted for clarity. **b** Map of interatomic vectors connecting equivalent α carbons in Phi and Pre-Chi. The length of each vector is proportional to the distance between the atoms in Phi and Pre-Chi. **c** The four main interfaces between the motors in Phi are highlighted in the context of the boxed model in panel (**a**): Linker:Linker, Linker:AAA4, AAA5:AAA5, and Stalk:Stalk. **d–g** Close-ups of the three interfaces highlighted in (**c**) that are disrupted by the formation of Pre-Chi: Linker:Linker (**d**), AAA5:AAA5 (**e**), and Linker:AAA4 (**f, g**). The top panel corresponds to the Phi model, and the bottom panel corresponds to the Pre-Chi model. Key residues, motor chains, LIS1 ring, and the domain(s) of the motor being displayed are highlighted on each panel. Interactions are shown with dotted lines, with their distances (in Å) indicated. There was no significant change in the Stalk:Stalk interface. There are differences in the Linker:AAA4 interface between the LIS1-bound face (**f**) and the LIS1-free face (**g**).

complexes, likely by relieving dynein autoinhibition in the Phi conformation.

## Discussion

### LIS1 binding to dynein is asymmetric in Pre-Chi

An interesting feature we observed in Pre-Chi was the asymmetric binding of LIS1 to the dynein tail (Fig. 2b). We confirmed that whenever we observed density for the IC-LC tower of the accessory chains in the Pre-Chi maps, LIS1 was bound to that same face. We hypothesize that the asymmetric nature of the Phi tail induces preferential breathing of the side where the bulk of the accessory chains are found, which in turn leads to the binding of LIS1 to that face. Given this, we speculate that this face might also be more labile during tail rearrangements downstream in the dynein-LIS1 activation pathway (Supplementary Fig. 8). Although several studies have shown that Nde1 and Ndel1, and their fungal homologs, co-localize with dynein and LIS1, and may be involved in recruiting LIS1 to dynein[48–51], our structural analysis shows that the asymmetric interaction of LIS1 with Phi dynein is driven by the properties of the Phi conformation itself and does not require additional factors.

### Two LIS1 dimers are likely required throughout the activation and assembly process

While we did not observe Chi in the cryo-EM data we collected for this study, Chi-specific interfaces are maintained in Pre-Chi (Fig. 4). The sample from which we previously solved Chi was designed to be monomeric (i.e., motor domains lacking the tail) and was trapped in a specific nucleotide state[27,33]. In that sample, only a small percentage of particles corresponded to the Chi conformation. We also observed a small percentage of Chi-like 2D class averages when we incubated monomeric dynein with ATP[37]. These observations from our previous studies suggest that Chi is a short-lived species in monomeric samples and could be an even shorter-lived species in a sample containing full-length dynein in the presence of ATP.

Several observations support the existence of Chi as an intermediate that follows Pre-Chi. First, the two LIS1 dimers wedged between the dynein motors in Chi disrupt more Phi-stabilizing interactions than Pre-Chi. Importantly, the second LIS1 dimer seen in Chi increases the distance between dynein's linkers; this would contribute to the destabilization of the tail that is required for the conformational changes need to convert its pseudo-twofold symmetry to the parallel

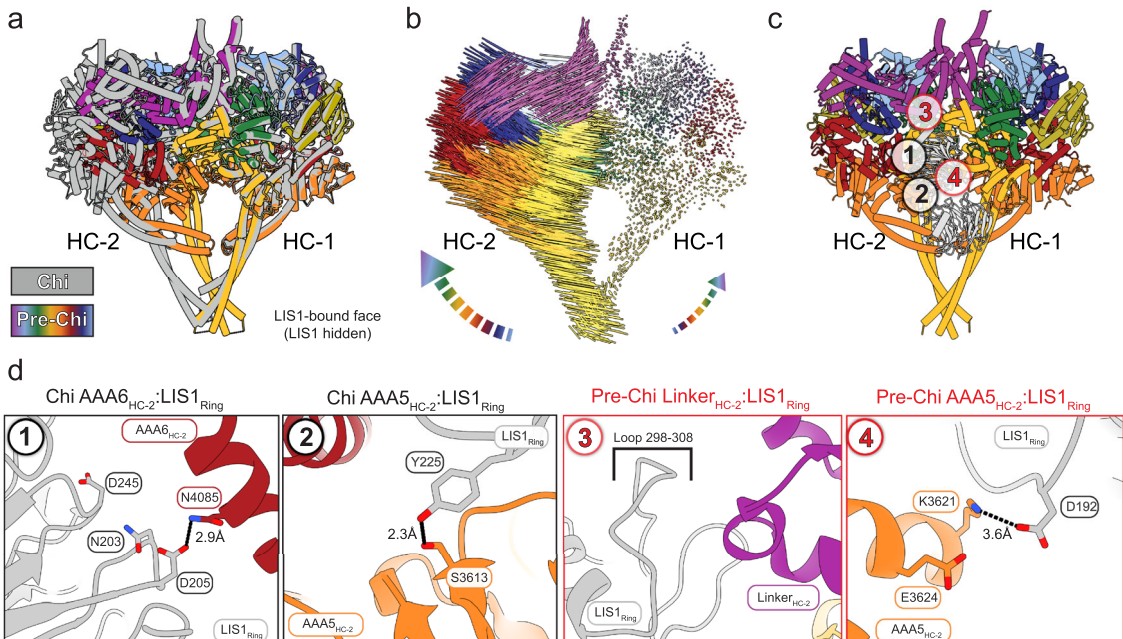

**Fig. 4 | Comparison of Pre-Chi with Chi. a** Superposition between Chi and Pre-Chi. The model of human Chi from our previous work (dark gray)[33] and human Pre-Chi motor (rainbow) were superimposed and aligned using HC-1. **b** Interatomic vectors connecting equivalent alpha carbons in Pre-Chi and Chi for the superposition shown in (**a**). **c** Pre-Chi model viewed from the LIS1-bound face with interfaces present in both Chi and Pre-Chi (#1-2 in black circles) and Pre-Chi-specific interfaces (#3-4 in red circles) highlighted. **d** Close-ups of the interfaces highlighted in (**c**). Residues involved in the interfaces and the names of the domains interacting with LIS1$_{ring}$ are highlighted in each panel. Interactions are shown with dotted lines, with their distances (in Å) indicated.

arrangement in the active DDA complex. Finally, DDA complexes have been shown, structurally, to contain two LIS1 dimers bound to a single dynein dimer[3], a stoichiometry consistent with Chi. These observations suggest that a stoichiometry of two LIS1 per dynein dimer may be present throughout the activation and assembly process.

### Model for dynein activation by LIS1

Determining how LIS1 facilitates the relief of dynein autoinhibition and the assembly of a fully activated DDA complex has been a major challenge in the field. Here, we used a heterogeneity mining approach to answer how the autoinhibition of full-length human dynein is relieved by LIS1 in the presence of ATP. We identified Pre-Chi as an intermediate state in the LIS1-mediated dynein activation pathway and propose that this species represents an early step in the relief of dynein autoinhibition. A single set of LIS1 β-propellers are wedged on one side between the dynein motor domains; this partially disrupts three of the four interfaces responsible for maintaining dynein autoinhibition (Fig. 5d, #1-2).

In this model, Pre-Chi is followed by Chi, where a second set of LIS1 β-propellers becomes wedged between the dynein motor domains, fully disrupting three of the four interfaces responsible for maintaining dynein autoinhibition (Fig. 5d, #3)[33]. This disruption likely initiates structural changes propagating throughout the complex, including the tail. While the specific role of the tail during this transition remains unclear, we hypothesize that either at the Chi stage, or at state(s) closely following Chi, the dynein complex undergoes a significant conformational shift. This shift would involve transitions from the twisted pseudo-twofold symmetry of the autoinhibited state to the parallel symmetry associated with the translational symmetry of the dynactin complex to which dynein must bind. Following the release of dynein autoinhibition, LIS1 facilitates the coordination of dynein and dynactin to assemble the Dynein-Dynactin-JIP3-LIS1 (DDA-LIS1) complex (Fig. 5d, #4)[3]. We hypothesize that LIS1-mediated coordination, either within this complex or in earlier intermediates, enhances the recruitment of activating adapters, thereby supporting the assembly

of motile dynein complexes. Once the DDA complex forms on microtubules, LIS1 disengages from the complex, allowing the active transport complex to move at full velocity (Fig. 5d, #5).

Although the structures of Pre-Chi and Chi dynein provide a pathway for LIS1 to relieve the Phi conformation, the LIS1-mediated interactions that form these intermediates must, in turn, be disrupted for dynein to be assembled into an active DDA complex. A possible mechanism is suggested by the observation that yeast LIS1 increases dynein's ATP hydrolysis rate in the absence of microtubules[37], which would increase the conformational flexibility of dynein, lowering the activation energy for the conformational changes required for full activation. Thus, after binding of LIS1 has led to the formation of Pre-Chi and Chi, an increase in the rate of ATP hydrolysis of dynein would facilitate the rearrangements that lead to the formation of the fully active DDA complex. Future analysis using all components of the DDA complex and LIS1, in the presence of ATP, could capture a broader range of intermediate states and provide a deeper understanding of how LIS1 facilitates the activation and assembly of dynein.

## Methods

### Cloning, plasmid construction, and mutagenesis

The plasmids for full-length human cytoplasmic-dynein 1 (Addgene plasmid # 111903) and human LIS1 (Addgene plasmid #132539) were gifts from Andrew Carter (LMB-MRC). Human LIS1$^{\Delta298-308}$ was generated through Genescript Express Mutagenesis & Site-Directed DNA Mutagenesis service.

### Protein expression and purification

For baculovirus generation, each construct was individually transformed into DH10EMBacY chemically competent cells (Geneva Biotech), and bacmid DNA from these constructs was extracted using a blue/white colony screen. Colony PCR was used to confirm each dynein chain prior to transfection into SF9 cells. The recombinant baculovirus for each construct was individually produced in 2 mL of SF9 cells at $0.5 \times 10^6$ cells/mL by transfection of the bacmid DNA using

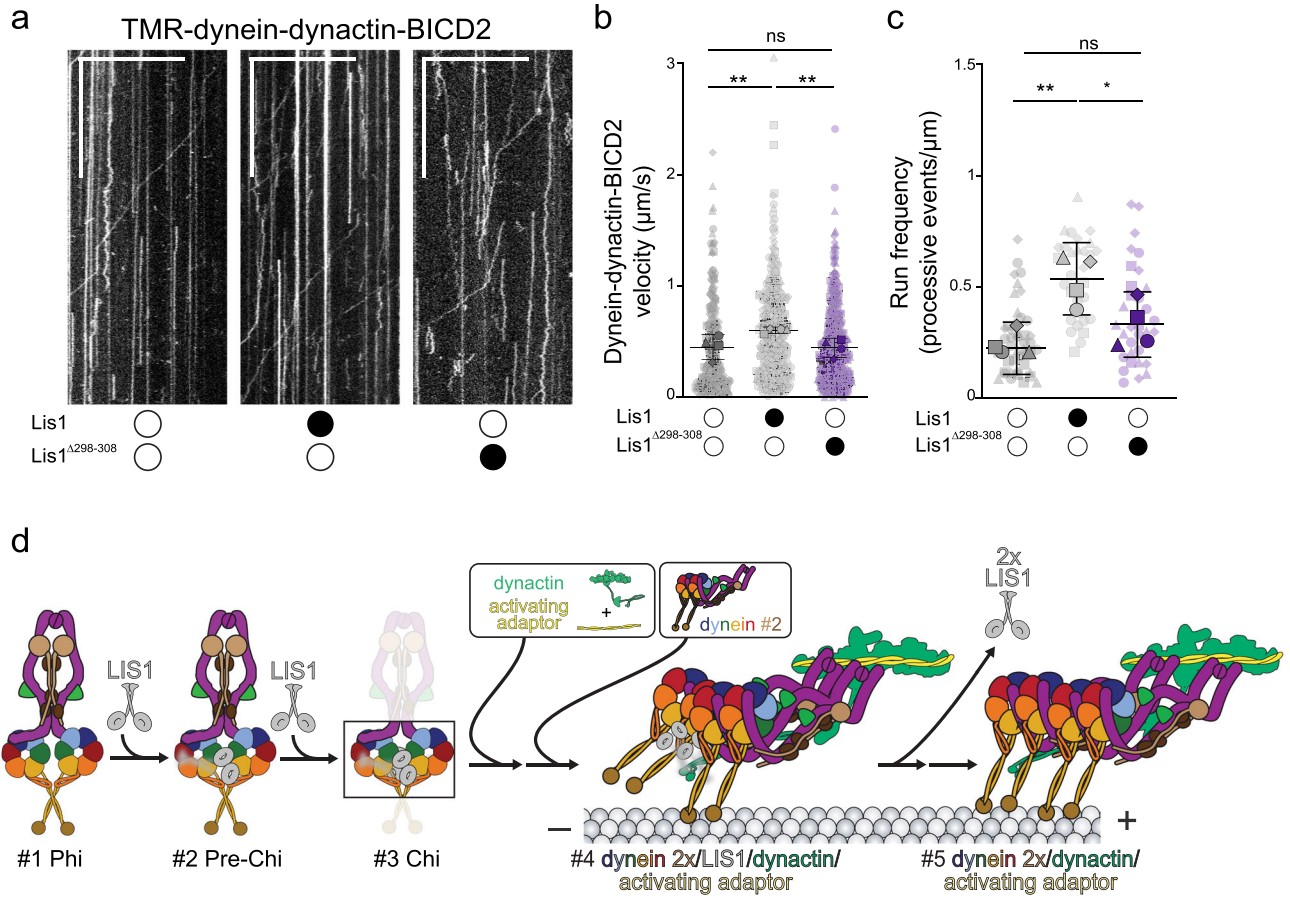

**Fig. 5 | The Pre-Chi-specific contact between LIS1 and dynein's linker is required for LIS1's effect on dynein motility. a** Representative kymographs from single-molecule motility assays with purified TMR−dynein−dynactin−BICD2 in the absence (white circle) or presence (black circle) of human LIS1 wild type or LIS1$^{\Delta298-308}$. Scale bars, 10 μm (x) and 100 s (y). **b** Single-molecule velocity (mean ± standard deviation of the means of each replicate) of TMR−dynein−dynactin−BICD2 complexes in the absence (white circles) or presence (black circles) of human LIS1 or LIS1$^{\Delta298-308}$. Superplots show all individual data points for each of the four technical replicates. n values for each replicate are: no LIS1, n = 70, 70, 50, 91; LIS1, n = 90, 85, 58, 117; LIS1$^{\Delta298-308}$, n = 116, 125, 89, 124. Larger shapes denote the mean of each of the four technical replicates. No LIS1 and LIS1 **P = 0.0017, No LIS1 and LIS1$^{\Delta298-308}$ ns P = 0.9692, LIS1 and LIS1$^{\Delta298-308}$ **P = 0.0012. Statistics were generated on the means of the four replicates using a One-Way ANOVA with Tukey's multiple comparison test. **c** Superplots show Run frequencies (processive events /μm of microtubule length; mean ± standard deviation of the means of each replicate) of TMR−dynein−dynactin−BICD2 complexes in the absence (white circle) or presence (black circle) of unlabeled wild-type human LIS1 or LIS1$^{\Delta298-308}$. Data points are represented as triangles, circles, squares, and diamonds corresponding to single measurements within each technical replicate (no LIS1, n = 14, 12, 11, 8; LIS1, n = 12, 7, 7, 7; LIS1$^{\Delta298-308}$, n = 11, 5, 8, 9). No LIS1 and LIS1 **P = 0.005, LIS1 and LIS1$^{\Delta298-308}$ *P = 0.0466, no LIS1 and LIS1$^{\Delta298-308}$ ns P = 0.3453. Statistical analysis was done using a One-Way ANOVA with Tukey's multiple comparison test. **d** Role of LIS1 in the activation of dynein. This schematic is an updated version of the pathway introduced in Fig. 1b that incorporates the Pre-Chi intermediate identified in this study. Source data for (**b** and **c**) are included with this manuscript, along with their corresponding statistical tests.

FuGene HD (Promega) according to the manufacturer's protocol (V0). After three days of incubation, V0s were harvested, and 1 mL of V0 was used to infect a 50 mL culture of SF9 at 1 × 10⁶ cells/mL (V1). After three days of incubation, V1 viruses were harvested and stored at 4 °C in the dark until required. Protein expression was induced by adding 6 mL of V1 to 600 mL of SF9 cells at 1 × 10⁶ cells/mL and incubating the cells at 27 °C on a shaker for three days at 90 rpm. After the third day, cells were harvested by centrifugation (2168 x g for 5 min at 4 °C), and the pellet was resuspended in ice-cold 1x PBS and pelleted again. The supernatant was discarded, and the pellet was flash-frozen in liquid nitrogen and stored at − 80 °C.

Purification of full-length human cytoplasmic-dynein 1, human LIS1, and human LIS1$^{\Delta298-308}$ was carried out as previously described[17,28,36,52,53]. Specifically, 4 × 600 mL of frozen insect cell pellet were used to purify full-length human cytoplasmic-dynein 1, and 2 × 600 mL of frozen insect cell pellet were used to purify human LIS1 and LIS1$^{\Delta298-308}$. For LIS1 and LIS1$^{\Delta298-308}$, both had a final concentration

of ~1 mg/mL and were flash-frozen in liquid nitrogen and stored at − 80 °C until use. For dynein, the protein was concentrated to ~ 0.1 mg/mL and used fresh to prepare cryo-EM grids the same day.

**Electron microscopy sample preparation**
Purified full-length dynein and LIS1 were combined at a 1:1 molar ratio with final concentrations of 0.3 μM. ATP was added to this sample to a final concentration of 1 mM. The sample was incubated at room temperature for 5 minutes. For plunge freezing, we used a custom manual plunge freezer (UCSD Cryo-EM Facility) located in a humidified (> 95% relative humidity) cold room maintained at 4 °C[54]. In the cold room, the sample was applied to an UltraAuFoil 1.2/1.3, 300 grid (Quantifoil) that had been plasma-cleaned using a Gatan Solarus II plasma cleaner (10 s, 15 Watts, 75% Ar/25% O2 atmosphere). The grid was manually blotted for ~ 4 to 5 s using Whatman No.1 filter paper and immediately vitrified in a 50:50 ethane:propane liquid mixture cooled by liquid nitrogen.

## Electron microscopy image collection and data processing

Data was collected at the UCSD Cryo-EM Facility on a Titan Krios G4 (Thermo Fischer Scientific) operating at 300 keV with a Falcon 4 direct electron detector and a Selectris X energy filter (<10 eV slit size). Automated data collection was performed using EPU, and images were collected at a nominal magnification of 130,000x (0.935 Å/pixel size) with a total exposure dose of ~55 electrons/Å$^2$. The defocus range was set to 0.5–2.5 µm.

On-the-fly motion correction and CTF estimation were done through cryoSPARC Live[39]. A total of 12,792 movies were collected initially, but we pruned that down to 11,686 movies by setting the parameters of CTF Fit to a range of 1–5 Å and Defocus Avg to a range of 0.61–3,25 µm in the Overview setting of cryoSPARC Live. Particles were initially selected using a blob picker in cryoSPARC Live, 2D classified, and the resulting averages were used as templates for Topaz training[42]. Particles were extracted after Topaz training with a box size of 352 pixels and binned to 7.48 Å/pixel, yielding 1,937,842 particles. These particles were subjected to two rounds of 2D classification, which yielded a final set of 690,084 particles. After 2D classification, the particles were used for ab initio reconstruction of six classes, yielding four different Open classes, one Phi class, and one Pre-Chi class. All six classes with particles and maps were used for Heterogeneous Refinement, where we obtained one Phi class, one Pre-Chi class, and two types of Open classes (Partial Open Bent/LIS1 and Open Straight/LIS1).

The particles corresponding to the Phi and Pre-Chi classes were re-extracted, unbinned to 0.935 Å/pixel. These particles and the corresponding maps were used to perform non-uniform refinement, followed by local refinement and CTF/Defocus refinement to obtain final maps of the Phi and Pre-Chi motor domains. The particles from Phi and Pre-Chi from heterogeneous refinement were taken and combined to process the tails of Phi and Pre-Chi. The box size was expanded to 640 pixels and binned to 2.34 Å/pixel, yielding 195,092 particles. These particles were used to obtain reconstructions of the tails, followed by homogenous refinement. The map and particles were then taken into cryoDRGN, down-sampled to 128 pixels (2.34 Å/pixel), and used for low-resolution cryDRGN training[43]. Particles and maps that corresponded to Phi and Pre-Chi were separated through K-means clustering. The clusters corresponding to Phi had 127,751 particles, and those for Pre-Chi had 45,549 particles. The particles were brought back into cryoSRPAC, the box size was expanded to 1200 pixels and binned to 9.35 Å/pixel, and these particles were used for a round of 2D classification followed by reconstruction and refinement. This yielded a low-resolution full-length dynein particle, which was used to recenter the particles to the center of the tail, with the box size decreased to 416 pixels and binned to 3.24 Å/pixel. These particles were used for another round of 2D classification and reconstruction. Finally, the particles were unbinned (0.935 Å/pixels), and a non-uniform refinement was used to generate the final map.

For the Partial Open Bent/LIS1, the particles and map from the heterogeneous refinement were taken and unbinned to 0.935 Å/pixel and used for refinement to generate a consensus map for cryoDRGN training. This map and particles were taken into cryoDRGN, down-sampled to 128 pixels, and binned to 2.57 Å/pixel. After cryoDRGN low-resolution training, the particles that had the best and unique clusters were separated and taken back into cryoSPARC for another round of ab-initio reconstructions and refinement.

For the Open Straight/LIS1, the particles and map from the heterogenous refinement were taken and unbinned to 0.935 Å/pixel and used for refinement to generate a consensus map for RELION 3D classification without alignment[45]. The best classes from here were taken back into cryoSPARC for another round of ab initio reconstructions and 3D refinement.

## Model building and refinement

CryoSPARC's sharpening tools were used to enhance the quality of all maps used for model building. For the motor domain maps, the initial PDB models are listed in Supplementary Table 1. These models were docked into the corresponding motor domain densities using ChimeraX and adjusted as needed[55]. These models were subjected to rounds of real-space refinement in Phenix and manual corrections in Coot[56,57].

LIS1 was placed by rigid body docking of the previously determined human dynein monomer bound to the 2 x LIS1 structure (PDB: 8DYU). The β-sheets of the LIS1 β-propeller fit well into the map, and a short α-helix (residues 288–297), the only α-helix within the β-propeller, also fits its corresponding density in both the Site$_{ring}$ and Site$_{stalk}$ locations. The orientation of this helix and the positioning of the binding interface were consistent with previous structures[46].

For the tail domains of the Phi and Pre-Chi species, the heavy chain was initially modeled based on PDB: 8PTK. Newly resolved densities in these species required further modeling, which was accomplished using AlphaFold2 to predict the newly resolved areas of the accessory chains, including the intermediate chain (IC), light intermediate chain (LIC), and light chain (LC)[58,59]. The predicted models were docked into the density maps with ChimeraX and refined through additional rounds of Phenix real-space refinement and manual adjustments in Coot.

## Single-molecule motility assays and TIRF microscopy imaging

Single-molecule motility assays were performed in flow chambers constructed using glass coverslips. To reduce non-specific binding, coverslips were washed in 1 M hydrochloric acid (HCl), heated to 55 °C, for at least 4 h. Cleaned coverslips were mounted on a slide with double-sided tape, creating ~10 µL volume chambers. Motility chambers were coated (5 min at room temperature) sequentially with 1 mg/mL Biotin-BSA (SIGMA), 0.5 mg/mL streptavidin (Thermo Scientific) diluted in dynein lysis buffer (DLB; 30 mM HEPES pH 7.4, 50 mM potassium acetate, 2 mM magnesium acetate, 1 mM EGTA, 10% glycerol, 1 mM DTT). Microtubules were polymerized from tubulin prepared from bovine brain, as previously described[53] and contained ~10% biotin-tubulin (Cytoskeleton) for attachment to the biotin-BSA-streptavidin coated chambers and ~10% Alexa Fluor 488 (Thermo Fisher Scientific) tubulin for visualization. Imaging was done in dynein-lysis buffer supplemented with 20 µM taxol, 1 mg mL$^{-1}$ casein, 5 mM Mg-ATP, 71.5 mM β-mercaptoethanol, and an oxygen scavenger system containing 0.4% glucose, 45 µg mL$^{-1}$ glucose catalase (Sigma-Aldrich), and 1.15 mg mL$^{-1}$ glucose oxidase (Sigma-Aldrich). Measurement of motility was done by imaging 0.125 nM labeled TMR-dynein, 0.9 nM unlabeled dynactin, 5 nM BICD2 complexes alone with LIS1 buffer (10 mM Tris pH 8.0, 2 mM MgOAc, 150 mM KOAc, 1 mM EGTA, 1 mM DTT, 10% glycerol), 300 nM LIS1 or LIS1$^{\Delta298-308}$. The dynein, dynactin, and BICD2 complexes were incubated on ice for 10 min to allow the dynein-dynactin-activator adapter complex to form. Subsequently, LIS1 or buffer was added to the active dynein complexes, and the mixture was incubated on ice for an additional 10 min prior to TIRF imaging. Three movies per condition were recorded every 0.3 s for 3 min.

Imaging was performed with an inverted microscope (Nikon, Ti-E Eclipse) equipped with a 100×1.49 N.A. oil immersion objective (Nikon, Plano Apo). The *xy* position of the stage was controlled by a ProScan linear motor stage controller (Prior). The microscope was equipped with an MLC400B laser launch (Agilent) with 405 nm (30 mW), 488 nm (90 mW), 561 nm (90 mW), and 640 nm (170 mW) laser lines. The excitation and emission paths were filtered using appropriate single bandpass filter cubes (Chroma). The emitted signals were detected with an electron multiplying CCD camera (Andor Technology, iXon Ultra 888). Illumination and image acquisition were controlled by NIS Elements Advanced Research software (Nikon).

## TIRF motility data analysis

TIRF motility data analysis was done on kymographs generated in Fiji[34,60]. Samples were blinded prior to kymograph generation.

Kymographs were generated for each microtubule in the three movies recorded for each condition. Velocities were calculated from molecules that moved processively (continuously moving along a microtubule track) for more than five frames. Non-motile or diffusive events were counted separately and were not included in the velocity analysis. Processive events were defined as events that move uni-directionally and do not exhibit directional changes greater than 600 nm. Diffusive events were defined as events that exhibit at least one bidirectional movement greater than 600 nm in each direction. Static events were defined as events that do not exhibit movement (less than 600 nm in each direction). Single-molecule movements that change apparent behavior (for example, shifting velocity mid-run or from non-motile to processive) were counted as multiple velocity events. Run frequency (processive events/ μm) were calculated by counting the number of processive events for each microtubule in individual movies and dividing this number by the microtubule length. Data was unblinded before proceeding to statistical analysis.

## Statistical analysis

Brightness and contrast were adjusted in Fiji for videos and kymographs. All statistical tests were generated using GraphPad Prism 10. The exact value of *n*, evaluation of statistical significance, *P*-values, and specific statistical analysis are described in the corresponding figures and figure legends. All TIRF experiments were analyzed from four independent replicates, and individual analysis of each replicate showed similar results. For velocity analysis, frequency distributions were first calculated for each replicate, and data were fit to a Gaussian distribution to calculate mean values. Statistical analysis was performed on the means of each biological replicate using a one-way ANOVA with Tukey's multiple comparison test. Source data are provided with this paper.

## Reporting summary

Further information on research design is available in the Nature Portfolio Reporting Summary linked to this article.

## Data availability

Cryo-EM maps have been deposited in the Electron Microscopy Data Bank under accession codes: Pre-Chi motor – EMD-47342, Pre-Chi tail – EMD-47430, Phi motor – EMD-47373, Phi tail – EMD-47443, Bent/2xLIS1 – EMD-47370, Bent – EMD-47360, Bent/1xLIS1 – EMD-47372, Interm. – EMD-47371, Straight/1xLIS1 – EMD-47429, Straight – EMD-47377. Atomic coordinates have been deposited in the Protein Data Bank under accession codes: Pre-Chi motor – 9DZY, Pre-Chi tail – 9E23, Phi motor – 9E0X, Phi tail – 9E28, Bent/2xLIS1 – 9E0T, Bent – 9E0K, Bent/1xLIS1 – 9E0W, Interm. – 9E0U, Straight/1xLIS1 – 9E22, Straight – 9E0Y. Unprocessed micrographs are deposited to EMPIAR under the following accession numbers: EMPIAR-12715. All other data will be made available upon request. Source data are provided with this paper. Source data and analyses for the distribution of particles corresponding to the three main species identified in the cryo-EM dataset (Fig. 1e, f) and dynein in vitro motility assays (Fig. 5b, c) are provided with this paper. Source data are provided in this paper.

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

## Acknowledgements

We thank the Cryo-EM Facility at UC San Diego. We thank Mark Herzik and Adam Fenton for useful discussions about the manuscript. We also thank our funding sources: NIH R01 GM107214 and R35 GM145296EPK (AL); and NIH R35 GM141825 and the Howard Hughes Medical Institute (S.L.R.-P.). K.H.V.N. was supported by the Molecular Biophysics Training Grant (NIH grant T32 GM139795); E.P.K. was supported by a Jane Coffin Childs Postdoctoral Fellowship; and A.A.K. was supported by NIH P30 CA014195.

## Author contributions

K.H.V.N. purified all proteins, prepared cryo-EM samples, collected and processed cryo-EM data, and built all atomic models; E.P.K. performed the single-molecule experiments; S.L.R.-P. and A.E.L. supervised the work; K.H.V.N., E.P.K., A.A.K., S.L.R.-P., and A.E.L. wrote and edited the manuscript.

## Competing interests

The authors declare no competing interests.
