## [Transparent Peer Review file · Nature Communications]

Cryo-EM captures early intermediate steps in dynein activation by LIS1

Corresponding Author: Dr Andres Leschziner

Version 0:

Reviewer comments:

Reviewer #1

(Remarks to the Author)

Nguyen et al. used cryo-EM to map the conformational landscape of dynein activation by LIS1, a known activator of this microtubule-based motor. During activation with LIS1, dynein undergoes a series of conformational changes, starting from the autoinhibited Phi state, progressing through a series of intermediate states, and culminating in the fully activated, force-generating state. In this final state, dynein forms a complex with dynactin and cargo-specific adaptors (DDA), and LIS1 is released. Previous work identified the activation intermediate Chi state, which suggested a mechanism for LIS1-mediated relief of autoinhibition by wedging between the β -propellers of dynein's motor domain. However, to trap the Chi state, these studies used non-native conditions, including a truncated, monomeric dynein construct, mutants, and non-hydrolysable ATP analogs.

The current study makes two major contributions:

1. It employs conditions that more closely mimic the native dynein activation environment, using dynein dimers, LIS1 dimers, and ATP, allowing the motor to sample a range of intermediate states in cryo-EM grids.
2. It leverages new cryo-EM methodologies to perform what they term heterogeneity mining, capturing less populated intermediate states. This approach led to the discovery of a previously uncharacterized intermediate, the Pre-Chi state, which they propose sits between the autoinhibited Phi state and the Chi state.
3. To test the functional relevance and in-solution existence of the Pre-Chi state, the authors introduced a LIS1 deletion mutant (LIS1 Δ 298-308) that disrupts dynein linker-LIS1 contacts specific to this state. This mutation impaired dynein motility in vitro, reinforcing the physiological significance of the findings.

This is a well-executed study, with strong evidence derived from careful classification of structural states using cryo-EM. The mutagenesis-motility experiments add substantial value by further validating the existence and functional relevance of the Pre-Chi state. The manuscript is clearly written and, in my opinion, merits publication in Nature Communications, requiring only minor revisions, mostly concerning the illustrations:

1. Fig. 1f is too small to effectively distinguish key states. The only discernible feature is whether LIS1 is present as a monomer, a dimer, or absent. The figure should be enlarged with clearer annotations added, such as arrows highlighting conformational differences. In Fig. 1e, the transition from Phi to Pre-Chi should be more prominently displayed for better clarity, and the text should be enlarged (e.g., "LIS1"). Additionally, ensure consistent nomenclature throughout the manuscript, as "LIS1" is sometimes written as "Lis1" (e.g., in Fig. 1f).
2. Figs. such as 3b and 4b do not clearly convey the nature of structural transitions: where do they begin and end? Is there a fulcrum for conformational changes? Including specific angular or translational values would also help quantify these transitions. If the goal is to illustrate movement, a more intuitive representation is needed in Fig. 4a-c.
3. Fig. 4d, intended to highlight the mutated loop, does not clearly illustrate how this loop's context differs in Chi and Pre-Chi states. A superimposition of these two states at a larger magnification would improve clarity and help better rationalize the effects of the deletion.
4. A movie morphing between structural states, pausing at each step to show the corresponding cryo-EM map, would significantly enhance clarity for readers.
5. The manuscript appears to have been initially formatted for another journal. Since Nature Communications uses Supplementary Figures rather than Extended Data Figures, this should be adjusted throughout.
6. Stylistic recommendation – the overuse of quotation marks (e.g., "Phi," "trapped," "tail," "intermediate," "chain") is unnecessary. Established nomenclatures should be presented without quotation marks.

Overall, this is a strong and insightful contribution that advances our understanding of the conformational transitions that take place during LIS1-mediated dynein activation.

Reviewer #2

(Remarks to the Author)

Dynein is responsible for most minus end directed transport along microtubules in cells. It is one of the most complex motor proteins and exists in an autoinhibited state termed the Phi-particle. Several factors contribute to its activation like the dynein activator dynactin, dynein cargo adaptors as well as the dynein regulator Lis1. The manuscript "Cryo-EM captures early intermediate steps in dynein activation by LIS1" by Nguyen et al. reports a new intermediate structural state during Lis1 mediated dynein motor complex activation.

The structural investigation of such transient states is currently a major challenge in the dynein field. The group recently reported the Chi-particle featuring two Lis1 dimers bound to the two dynein motor domains of the dynein complex. The presence of Lis1 disrupts the contacts responsible for Phi-particle stabilization, which primes the dynein motor complex for activation. In this study, the authors have used a state-of-the-art structural heterogeneity mining approach to investigate the dynein complex in the presence of ATP and Lis1. They were able to identify a "Pre-Chi" state of the human dynein complex, which features a single Lis1 dimer bound to one of the dynein motor domains of the dynein complex. Compared to the Chi-particle, fewer Phi-particle stabilizing contacts are disrupted. They are able to identify preChi-particle specific binding interfaces between a Lis1 loop region (aa298-308) and the dynein linker domain as well as Lis1 D192 and the AAA5 domain of the dynein motor ring. Deleting the Lis1 aa298-308 loop region abolishes the activating effect of Lis1 on dynein complex velocity and processivity, which clearly supports the functional relevance of the newly identified preChi-particle. The authors have carried out an impressive amount of work and the manuscript is well written and easy to follow. All experiments have been carefully carried out and the conclusions drawn from these experiments are justified. I recommend publication in Nature Communications. I just have a few minor issues as outlined below:

- page 3, line 47: dot is missing after "humans"
- page 11, line 236: replace "list here" with the nucleotide state
- The discussion might benefit from a few lines putting the proposed Lis1 activation model into context with the recently published Lis1 activation model for the Dynein-Dynactin-JIP3-Lis1 complex (Singh, Lau et al, 2024). It seems that Lis1 plays at least two roles during dynein motility activation. In the context of the preChi and Chi particles the Phi-particle stabilizing interactions are interrupted. When the dynein complex subsequently binds to dynactin, Lis1 promotes productive cargo adaptor binding to facilitate the formation of motile dynein complexes.

Reviewer #3

(Remarks to the Author)

In their manuscript, Nguyen et al. describe novel cryo-EM structures of cytoplasmic dynein-1 in various activation states, including a conformation that had not previously been described. To achieve this, they incubated a complex of dynein and its interactor LIS1 with ATP. After incubation, they vitrified the samples and collected the data for structure elucidation by cryo-EM. After 2D classification, particles were reconstructed into six different classes, allowing the elucidation of the structures in different activation states. Mutational analyses were also carried out, to complement the structural findings. The manuscript is very well written and clear. The identification of a new intermediate, and the approach of analyzing a conformational landscape, rather than trapping individual intermediates, are important findings that should be interesting to a broad audience.

Nevertheless, a few questions came up that could be further clarified in the manuscript:

1) The electron density and fit of the motor domains in the structures are mostly very good based on the provided models and maps. The bent linker-2xLIS1 map is well defined also in the LIS1 region, especially chain B, but chain C is still clearly defined. In the other structures containing LIS1, the LIS1 density is weaker and less defined than the dynein part. The best definition is found in the motor domains of the dynein.

In the bent linker-1xLIS1 structure, the density of the motor domains is clearly defined at a map level of 4 sigma (in Coot), while scrolling to ca. 2 sigma is needed to visualize the LIS1 density.

It would be helpful if you could shortly discuss the meaning of this. Possibly, there was some flexibility in the binding interface, and the alignment of the particles was driven by the core of dynein. Alternatively, there could be a partial occupancy of LIS1. If so, does this have any impact on the proposed model?

How did you proceed to unambiguously place the model of LIS1? Was the overall shape / connectivity of the density clear enough to place the LIS1 model, or was it necessary to use prior knowledge about the binding mode (for example from the better-defined structure with two LIS1 molecules bound) to place it?

2) The complexes were incubated with 1 mM ATP for 5 minutes prior to vitrification. For a better understanding of the experiment, it would be helpful if you could shortly explain how you chose the ATP incubation time and concentration. Would you expect to see different or additional intermediates with different ATP concentrations or incubation times? Is the ATP concentration high enough so that it is not used up during the incubation time? Considering that different states, and even a new state were identified, it is clear that the concept worked. But it would be helpful to have an understanding on how you decided on these conditions/incubation times.

3) In Figure 1 e/f, it seems that for dimers that appear in the micrographs, each monomer was boxed and separately analyzed. Is this correct? If so, do the two monomers belonging to a dimer have some correlation, for example, if one is bent,

does the other one also tend to be bent, or are they randomly distributed?

4) In Figure 1b, the labeling may be slightly confusing, with states 1-4 being circled in the Figure, shown in parentheses in the Figure legend, and indicated as # in the text (lines 87-89).

5) Are there no subclasses at all that have a different Lis1-binding stoichiometry in Pre-Chi?

6) Where in Extended Data Figure 7 is the labile face during tail rearrangement (suggested in lines 222-225 of the manuscript) obvious in panels b to e?

7) There is a typo on p. 10, line 218: "assymetric" should be with one "s" but two "m".

These questions can likely be addressed with minor corrections.

Reviewer #4

(Remarks to the Author)

The manuscript identifies new Lis1-DDA structures and proposes a stepwise activation pathway for dynein complexes. The 'Chi' inhibited conformation has been established, and this novel structure is between that and fully activated complexes. This is important work because the activation mechanism of dynein is not well understood and is very important for its intracellular function.

We reviewed the functional motility data in Figure 5, and I will leave comments on the structural work to the other reviewers. The velocity effects aren't large, but the velocity is pretty clear - you can see the shift in the distribution clearly by eye. Likewise the event frequency.

Overall the motility assays were carefully done and well described. A few comments:

- 1) The section title "The Pre-Chi-specific interface is required for human dynein complex assembly" is a little strong since you can get assembly without Lis1. Perhaps "...for efficient human dynein complex assembly".
- 2) "Run frequency" is a cleaner way to say processive events/um, then processive events/um are units.
- 3) Line 417-418. "Three images per condition were recorded every 0.3 s for 3 min." I'm not exactly sure what this is saying, do you mean movies, kymographs, or other? Reword to clarify.
- 4) There needs to be more description of how some events were treated. For example, how do you treat a run where the motor changes velocity without pausing? Is it two separate events or do you average the velocity for the whole run?

Reviewer #5

(Remarks to the Author)

REVIEWER COMMENTS

Reviewer #1 (Remarks to the Author):

Nguyen et al. used cryo-EM to map the conformational landscape of dynein activation by LIS1, a known activator of this microtubule-based motor. During activation with LIS1, dynein undergoes a series of conformational changes, starting from the autoinhibited Phi state, progressing through a series of intermediate states, and culminating in the fully activated, force-generating state. In this final state, dynein forms a complex with dynactin and cargo-specific adaptors (DDA), and LIS1 is released. Previous work identified the activation intermediate Chi state, which suggested a mechanism for LIS1-mediated relief of autoinhibition by wedging between the β -propellers of dynein's motor domain. However, to trap the Chi state, these studies used non-native conditions, including a truncated, monomeric dynein construct, mutants, and non-hydrolysable ATP analogs.

The current study makes two major contributions:

1. It employs conditions that more closely mimic the native dynein activation environment, using dynein dimers, LIS1 dimers, and ATP, allowing the motor to sample a range of intermediate states in cryo-EM grids.
2. It leverages new cryo-EM methodologies to perform what they term heterogeneity mining, capturing less populated intermediate states. This approach led to the discovery of a previously uncharacterized intermediate, the Pre-Chi state, which they propose sits between the autoinhibited Phi state and the Chi state.
3. To test the functional relevance and in-solution existence of the Pre-Chi state, the authors introduced a LIS1 deletion mutant (LIS1 Δ 298-308) that disrupts dynein linker–LIS1 contacts specific to this state. This mutation impaired dynein motility in vitro, reinforcing the physiological significance of the findings.

This is a well-executed study, with strong evidence derived from careful classification of structural states using cryo-EM. The mutagenesis-motility experiments add substantial value by further validating the existence and functional relevance of the Pre-Chi state. The manuscript is clearly written and, in my opinion, merits publication in Nature Communications, requiring only minor revisions, mostly concerning the illustrations:

We thank the reviewer for their thoughtful and positive assessment of our work. We appreciate the reviewer's support for publication and have implemented the suggested revisions to the illustrations as outlined below.

1. Fig. 1f is too small to effectively distinguish key states. The only discernible feature is whether LIS1 is present as a monomer, a dimer, or absent. The figure should be enlarged with clearer annotations added, such as arrows highlighting conformational differences. In Fig. 1e, the transition from Phi to Pre-Chi should be more prominently displayed for better clarity, and the text should be enlarged (e.g., "LIS1"). Additionally, ensure consistent nomenclature throughout the manuscript, as "LIS1" is sometimes written as "Lis1" (e.g., in Fig. 1f).

Due to space constraints, we were limited as to how much we could enlarge Figure 1f, but we did our best to maximize this panel. In addition, to improve clarity we added a small cartoon next to each volume to illustrate the main features of each conformation. In Figure 1e, we enlarged the text labels as suggested and bolded “LIS1”. To further help readers visualize the conformational changes we observed in our structures, we created an animation showing the transitions (as morphs) between key states, now included as Supplementary Movie 1. We thank the reviewer for pointing out the inconsistencies in the LIS1 nomenclature, which we have now corrected throughout the manuscript and figures.

2. Figs. such as 3b and 4b do not clearly convey the nature of structural transitions: where do they begin and end? Is there a fulcrum for conformational changes? Including specific angular or translational values would also help quantify these transitions. If the goal is to illustrate movement, a more intuitive representation is needed in Fig. 4a–c.

We thank the reviewer for bringing this up. After discussing it, we felt that nothing conveys structural transitions (motion) better than movies, so we decided to add those to illustrate these points. We created supplementary movies for Figure 3 (Supplementary Movies 2 and 3) and Figure 4 (Supplementary Movies 4 and 5). These movies show the conformational changes and help convey the directionality and extent of movement between states and interfaces.

3. Fig. 4d, intended to highlight the mutated loop, does not clearly illustrate how this loop's context differs in Chi and Pre-Chi states. A superimposition of these two states at a larger magnification would improve clarity and help better rationalize the effects of the deletion.

The predicted human Chi model we used in this manuscript was based on the Chi structure we previously solved using the yeast system (Karasmanis et al., 2023), as there is currently no experimentally determined structure of human Chi. For the human Pre-Chi structure, we modeled in the flexible loop based on low-resolution density present in that region. This loop is not resolved in any existing dynein-LIS1 structures and is predicted with very low confidence in AlphaFold, consistent with the high flexibility we observed. This is why it is not possible to show a superimposition of these two states as it would imply a certainty about their structures that does not yet exist.

4. A movie morphing between structural states, pausing at each step to show the corresponding cryo-EM map, would significantly enhance clarity for readers.

We thank the reviewer for the suggestion. This is now Supplementary Movie 1.

5. The manuscript appears to have been initially formatted for another journal. Since Nature Communications uses Supplementary Figures rather than Extended Data Figures, this should be adjusted throughout.

Indeed, the manuscript was transferred directly from *Nature Structural and Molecular Biology* to *Nature Communications*, without reformatting. We have edited the manuscript in accordance with *Nature Communications* formatting guidelines.

6. Stylistic recommendation – the overuse of quotation marks (e.g., “Phi,” “trapped,” “tail,” “intermediate,” “chain”) is unnecessary. Established nomenclatures should be presented without quotation marks.

We thank the reviewer for this suggestion. We have revised the manuscript to remove unnecessary quotation marks as recommended.

Overall, this is a strong and insightful contribution that advances our understanding of the conformational transitions that take place during LIS1-mediated dynein activation.

We thank the reviewer for the positive and thoughtful feedback.

Reviewer #2 (Remarks to the Author):

Dynein is responsible for most minus end directed transport along microtubules in cells. It is one of the most complex motor proteins and exists in an autoinhibited state termed the Phi-particle. Several factors contribute to its activation like the dynein activator dynactin, dynein cargo adaptors as well as the dynein regulator Lis1. The manuscript “Cryo-EM captures early intermediate steps in dynein activation by LIS1” by Nguyen et al. reports a new intermediate structural state during Lis1 mediated dynein motor complex activation.

The structural investigation of such transient states is currently a major challenge in the dynein field. The group recently reported the Chi-particle featuring two Lis1 dimers bound to the two dynein motor domains of the dynein complex. The presence of Lis1 disrupts the contacts responsible for Phi-particle stabilization, which primes the dynein motor complex for activation. In this study, the authors have used a state-of-the-art structural heterogeneity mining approach to investigate the dynein complex in the presence of ATP and Lis1. They were able to identify a “Pre-Chi” state of the human dynein complex, which features a single Lis1 dimer bound to one of the dynein motor domains of the dynein complex. Compared to the Chi-particle, fewer Phi-particle stabilizing contacts are disrupted. They are able to identify preChi-particle specific binding interfaces between a Lis1 loop region (aa298-308) and the dynein linker domain as well as Lis1 D192 and the AAA5 domain of the dynein motor ring. Deleting the Lis1 aa298-308 loop region abolishes the activating effect of Lis1 on dynein complex velocity and processivity, which clearly supports the functional relevance of the newly identified preChi-particle. The authors have carried out an impressive amount of work and the manuscript is well written and easy to follow. All experiments have been carefully carried out and the conclusions drawn from these experiments are justified. I recommend publication in *Nature Communications*. I just have a few minor issues as outlined below:

We thank the reviewer for their positive comments.

- page 3, line 47: dot is missing after "humans"

Thank you for pointing this out; the missing period has been added.

-page 11, line 236: replace "list here" with the nucleotide state

We thank the reviewer for catching this typo. "List here" was an unintended placeholder and has been removed; this sentence was not meant to include a list of nucleotide states.

-The discussion might benefit from a few lines putting the proposed Lis1 activation model into context with the recently published Lis1 activation model for the Dynein-Dynactin-JIP3-Lis1 complex (Singh, Lau et al, 2024). It seems that Lis1 plays at least two roles during dynein motility activation. In the context of the preChi and Chi particles the Phi-particle stabilizing interactions are interrupted. When the dynein complex subsequently binds to dynactin, Lis1 promotes productive cargo adaptor binding to facilitate the formation of motile dynein complexes.

We apologize for the poor wording in our original manuscript. When we used "DDA-LIS1", we were indeed referring to the Dynein-Dynactin-JIP3-LIS1 complex the reviewer brought up, and cited the corresponding paper (Singh, Lau et al., 2024), but this was obviously not clear enough. We also agree with the proposed model that LIS1 plays multiple roles during dynein activation, including promoting productive cargo adaptor binding following relief of autoinhibition. To improve clarity, we have now explicitly defined this complex (DDA-LIS1) in both the Introduction and the Discussion. In the Introduction (lines 87–90), we state: "*LIS1 promotes the assembly of the Dynein-Dynactin-JIP3-LIS1 (DDA-LIS1) complex by binding to the p150 subunit of dynactin via its LIS1-N domain, while simultaneously maintaining interactions with dynein (Fig. 1b, #3 asterisk)³.*" In the Discussion (lines 279–283), we now include: "*Following the release of dynein autoinhibition, LIS1 facilitates the coordination of dynein and dynactin to assemble the Dynein-Dynactin-JIP3-LIS1 (DDA-LIS1) complex (Fig. 5d, #4)³. We hypothesize that LIS1-mediated coordination, either within this complex or in earlier intermediates, enhances the recruitment of activating adaptors, thereby supporting the assembly of motile dynein complexes.*"

Reviewer #3 (Remarks to the Author):

In their manuscript, Nguyen et al. describe novel cryo-EM structures of cytoplasmic dynein-1 in various activation states, including a conformation that had not previously been described. To achieve this, they incubated a complex of dynein and its interactor LIS1 with ATP. After incubation, they vitrified the samples and collected the data for structure elucidation by cryo-EM. After 2D classification, particles were reconstructed into six different classes, allowing the elucidation of the structures in different activation

states. Mutational analyses were also carried out, to complement the structural findings.

The manuscript is very well written and clear. The identification of a new intermediate, and the approach of analyzing a conformational landscape, rather than trapping individual intermediates, are important findings that should be interesting to a broad audience.

We thank the reviewer for the positive feedback.

Nevertheless, a few questions came up that could be further clarified in the manuscript:

1) The electron density and fit of the motor domains in the structures are mostly very good based on the provided models and maps. The bent linker-2xLIS1 map is well defined also in the LIS1 region, especially chain B, but chain C is still clearly defined. In the other structures containing LIS1, the LIS1 density is weaker and less defined than the dynein part. The best definition is found in the motor domains of the dynein. In the bent linker-1xLIS1 structure, the density of the motor domains is clearly defined at a map level of 4 sigma (in Coot), while scrolling to ca. 2 sigma is needed to visualize the LIS1 density.

It would be helpful if you could shortly discuss the meaning of this. Possibly, there was some flexibility in the binding interface, and the alignment of the particles was driven by the core of dynein. Alternatively, there could be a partial occupancy of LIS1. If so, does this have any impact on the proposed model?

We thank the reviewer for raising these issues.

We do not believe flexibility at the LIS1 binding interface is the reason for the weaker density as all dynein-LIS1 structures solved to date have shown the same binding interface. Instead, the most likely explanation is, as the reviewer mentioned, the fact that the alignment of particles in our reconstruction is dominated by the dynein core. As is generally the case in cryo-EM, peripheral features pay a larger price (in resolution and quality of the density) for errors in rotational alignment (the error in position roughly scales linearly with the distance of the feature from the center of rotation). Given dynein's conformational flexibility, it is almost certain that any structure we solved is not 100% conformationally homogeneous. While small conformational differences will not have a large effect on the resolution of the motor domain itself (except at the very edges), they will change the position of LIS1, which is furthest from the center. This would lead to a weaker density, even if the dynein-LIS1 interface itself remains the same. The fact that both structures (bent linker-1xLIS1 and bent linker-2xLIS1) contained a relatively low number of particles (~25k) may have also contributed to lower local resolution and weaker density in the LIS1 region (Supplementary Fig. 2a).

Weaker density for LIS1_{Stalk} is most likely due to partial occupancy, as we have shown before that LIS1 can bind to Site_{Ring} alone, to both Site_{Ring} and Site_{Stalk}, but not to Site_{Stalk} alone, an observation we have interpreted as indicating that Site_{Stalk} is a weaker

site that requires the cooperativity from Site_{Ring} to be engaged. To clarify this point, we have added the following sentence to the main text where dynein-LIS1 interfaces are defined (lines 94–97): *“In monomeric dynein, LIS1 has been observed bound to both site_{ring} and site_{stalk}^{24,35–39}, and to site_{ring} in the absence of binding to site_{stalk}^{27,29,36–39}, but not to site_{stalk} only. This is consistent with a model where site_{stalk} binding is weaker and depends on cooperativity with site_{ring}.”*

These considerations do not affect our proposed model, which only requires binding of a LIS1 β -propeller at Site_{ring}.

How did you proceed to unambiguously place the model of LIS1? Was the overall shape / connectivity of the density clear enough to place the LIS1 model, or was it necessary to use prior knowledge about the binding mode (for example from the better-defined structure with two LIS1 molecules bound) to place it?

The placement of LIS1 in both the Bent 2xLIS1 and Bent 1xLIS1 reconstructions was guided by our previously determined structure of the human dynein monomer bound to 2xLIS1 (PDB: 8DYU; Reimer et al., 2023). While the density for LIS1 is not as well resolved as the dynein core, we were able to unambiguously place LIS1 into the map because all its β -sheets fit well into the density. Critically, there is a short α -helix in the LIS1 β -propeller (residues 288–297), that acts as a landmark during docking and model building. We could see the corresponding density, which the helix fit well, when docking LIS1 into Site_{ring} or Site_{stalk}.

The orientation of this α -helix and the placement of the LIS1 binding interface are also consistent with those observed in our previous studies (Reimer et al., 2023), further supporting the accuracy of the model building. To clarify how LIS1 was placed in our maps, we added a detailed description in the *Methods* section under *Model building and refinement* (lines 398–403): *“LIS1 was placed by rigid body docking of the previously determined human dynein monomer bound to 2xLIS1 structure (PDB: 8DYU). The β -sheets of the LIS1 β -propeller fit well into the map, and a short α -helix (residues 288–297), the only α -helix within the β -propeller, also fit its corresponding density in both the Site_{ring} and Site_{stalk} locations. The orientation of this helix and the positioning of the binding interface were consistent with previous structures⁴⁶.”*

2) The complexes were incubated with 1 mM ATP for 5 minutes prior to vitrification. For a better understanding of the experiment, it would be helpful if you could shortly explain how you chose the ATP incubation time and concentration. Would you expect to see different or additional intermediates with different ATP concentrations or incubation times? Is the ATP concentration high enough so that it is not used up during the incubation time? Considering that different states, and even a new state were identified, it is clear that the concept worked. But it would be helpful to have an understanding on how you decided on these conditions/incubation times.

The ATP concentration and incubation time used in our study (1mM ATP for 5 minutes) were chosen based on conditions used in our recent work (Kendrick et al, bioRxiv

2024), where multiple conformational states were observed in dynein and dynein-LIS1 complexes. While we did not perform an exhaustive optimization across time points and ATP concentration for this study, we adopted these conditions as they were effective in revealing structural heterogeneity in a related system (yeast dynein and LIS1) (Kendrick et al., bioRxiv 2024). In the same study, ATP hydrolysis measurements showed that 1mM ATP would not be depleted over the course of the incubation.

3) In Figure 1 e/f, it seems that for dimers that appear in the micrographs, each monomer was boxed and separately analyzed. Is this correct? If so, do the two monomers belonging to a dimer have some correlation, for example, if one is bent, does the other one also tend to be bent, or are they randomly distributed?

Thanks for raising this point. All dyneins in our micrographs are dimers: Phi, Pre-Chi, and Open. In the case of Phi and Pre-Chi, the dimeric nature of the molecule is obvious, in that the two motor domains interact with each other. It is in the Open dimers where the two motors were boxed and treated separately.

The question of whether correlations exist between the two motors in an Open dimer is a very interesting one, but technically challenging to address. The problem is ensuring that two given motor domains belong to the same dimer during the analysis. Since the tail domains are very flexible in the Open species (and thus “lost” during processing), this would have to be determined using the distance between motor domains. However, because of the flexibility in dynein, the length of the tail, and the particle density in our micrographs, there will be many instances when a motor domain is closer to another motor domain from a different dimer than to its true dimer mate. This could easily lead to correlations (or lack thereof) that are not a true reflection of the data.

4) In Figure 1b, the labeling may be slightly confusing, with states 1-4 being circled in the Figure, shown in parentheses in the Figure legend, and indicated as # in the text (lines 87-89).

We thank the reviewer for pointing this out. We have updated the figure and figure legend to match the text and now use the “#” notation consistently throughout to label states 1–4.

5) Are there no subclasses at all that have a different Lis1-binding stoichiometry in Pre-Chi?

We attempted to further subclassify the Pre-Chi particles using cryoDRGN (Supplementary Fig. 7), with the goal of identifying other states with a different LIS1 binding stoichiometry. While this analysis did not reveal additional species, we cannot rule out the presence of a minor subpopulation in which only one LIS1 β -propeller is bound given the limited number of Pre-Chi particles. Thus, subpopulations within Pre-Chi remain a possibility that may be resolved with larger datasets.

6) Where in Extended Data Figure 7 is the labile face during tail rearrangement (suggested in lines 222-225 of the manuscript) obvious in panels b to e?

In Supplementary Figure 7b–e, the top panel in each pair shows the face of the IC/LC tower, which corresponds to the labile face of dynein during tail rearrangement. This is also the same face to which LIS1 preferentially binds. For clarity, we have updated the figure legend to state: *“b–e. Volumes from clusters 2–5 shown in two views: front view of the IC/LC tower face (top row) and side view (bottom row). The IC/LC tower (labeled) is used as a reference to determine which side of Phi LIS1 is bound to.”*

7) There is a typo on p. 10, line 218: “assymmetric” should be with one “s” but two “m”.

We thank the reviewer for pointing out this typo, which we have corrected.

These questions can likely be addressed with minor corrections.

Reviewer #4 (Remarks to the Author):

The manuscript identifies new Lis1-DDA structures and proposes a stepwise activation pathway for dynein complexes. The 'Chi' inhibited conformation has been established, and this novel structure is between that and fully activated complexes. This is important work because the activation mechanism of dynein is not well understood and is very important for its intracellular function.

We reviewed the functional motility data in Figure 5, and I will leave comments on the structural work to the other reviewers.

The velocity effects aren't large, but the velocity is pretty clear - you can see the shift in the distribution clearly by eye. Likewise the event frequency.

Overall the motility assays were carefully done and well described.

We thank the reviewer for recognizing the importance of this work and for their careful consideration of our motility data.

A few comments:

1) The section title "The Pre-Chi-specific interface is required for human dynein complex assembly" is a little strong since you can get assembly without Lis1. Perhaps "...for efficient human dynein complex assembly".

We thank the reviewer for their suggestion. We have renamed our section as suggested.

2) “Run frequency” is a cleaner way to say processive events/um, then processive events/um are units.

As suggested, we have changed Figure 5c, the Figure 5 legend and the relevant text in the manuscript to relabel “processivity” as “run frequency”.

3) Line 417-418. “Three images per condition were recorded every 0.3 s for 3 min.” I’m not exactly sure what this is saying, do you mean movies, kymographs, or other? Reword to clarify.

We thank the reviewer for catching this ambiguity. This sentence refers to the three movies acquired per condition. Kymographs were generated from all microtubules included in these 3 movies. We have now clarified this information in our “Single-molecule motility assays and TIRF microscopy imaging” and “TIRF motility data analysis” section of our materials and methods.

4) There needs to be more description of how some events were treated. For example, how do you treat a run where the motor changes velocity without pausing? Is it two separate events or do you average the velocity for the whole run?

For our analyses, single-molecule movements that changed apparent behavior were considered as multi-velocity events and counted as multiple events. This includes runs where the velocity shifts mid run. We have now clarified this point in the statistical analysis section of our materials and methods.

We chose this approach to be consistent with previous publications (Karasmanis et al. 2023, Christiansen, Kendrick et al 2021).

Note that using average velocities for the whole run still produces the same statistically significant shifts between the means of replicates of the different conditions. In this case, the n values and numerical value of the mean velocity is slightly reduced across all conditions.

Reviewer #5 (Remarks to the Author):

I co-reviewed this manuscript with one of the reviewers who provided the listed reports. This is part of the NatureCommunications initiative to facilitate training in peer review and to provide appropriate recognition for Early Career Researchers who co-review manuscripts.

We thank the reviewer for their contribution.